# Toxicological Evaluation of Silver Nanoparticles Synthesized with Peel Extract of *Stenocereus queretaroensis*

**DOI:** 10.3390/ma15165700

**Published:** 2022-08-18

**Authors:** Eduardo Padilla-Camberos, Karen J. Juárez-Navarro, Ivan Moises Sanchez-Hernandez, Omar Ricardo Torres-Gonzalez, Jose Miguel Flores-Fernandez

**Affiliations:** 1Unit of Medical and Pharmaceutical Biotechnology, Center for Research and Assistance in Technology and Design of the State of Jalisco, A.C. (CIATEJ), Normalistas 800, Guadalajara 44270, Jalisco, Mexico; 2Department of Biochemistry & Centre for Prions and Protein Folding Diseases, University of Alberta, 204 Brain and Aging Research Building, Edmonton, AB T6G 2M8, Canada; 3Department of Research and Innovation, Universidad Tecnológica de Oriental, de la No. 3402, Calle 37 Nte., Oriental 75020, Puebla, Mexico

**Keywords:** silver nanoparticles, toxicity, genotoxicity, irritation, cytotoxicity

## Abstract

Silver nanoparticles (AgNPs) synthesized with plants are widely used in different industries, such as the medical, industrial, and food industries; however, their hazards and risks remain unclear. Here, we aimed to evaluate the toxicological effects of AgNPs in both in vitro and in vivo models. Previously, we developed and characterized green synthesized AgNPs based on *Stenocereus queretaroensis* (*S. queretaroensis*). The present study evaluates the toxicity of these AgNPs through cytotoxicity and mutagenicity tests in vitro, as well as genotoxicity tests, including the evaluation of acute oral, dermal, and inhalation toxicity, along with dermal and ocular irritation, in vivo, according to guidelines of The Organization for Economic Co-operation and Development (OECD). We evaluated cell cytotoxicity in L929 cells, and the half-maximal inhibitory concentration was 134.76 µg/mL. AgNPs did not cause genotoxic or mutagenic effects. Furthermore, in vivo oral, dermal, and acute inhalation toxicity results did not show any adverse effects or mortality in the test animals, and after the dermal and ocular irritation assessments, the in vivo models did not exhibit irritation or corrosion. Therefore, the results show that these previously synthesized *S. queretaroensis* AgNPs do not represent a risk at the tested concentrations; however, little is known about the effects that AgNPs induce on physiological systems or the possible risk following long-term exposure.

## 1. Introduction

The synthesis of nanoparticles is considered an emerging improvement and development tool in various industrial, food, medical, diagnostics, electronics, and aeronautical applications. Notably, the interest in and production of silver nanoparticles have exponentially grown due to their possible applications in the medical, cosmetic, and food industries due to their physicochemical properties [1,2].

There are many ways to synthesize AgNPs through different chemical, physical, and biological methods. Physical and chemical methods are expensive and not environmentally friendly, as their production requires toxic chemicals and releases toxic waste products to the environment. The green synthesis of AgNPs stands out as an environmentally friendly method, as it generates nanoparticles using plant extracts, with a lesser impact in the environment; therefore, it is an emerging approach to produce cheaper and less harmful nanoparticles. Moreover, plants are a good resource for reducing agents to synthesized AgNPs because of their alkaloids, saponins, tannins, phenols, and terpenoids contents [2,3,4,5]. Green AgNPs synthesis is an efficient, cheap, and cost-effective method which has led to a remarkable increase in the production of AgNPs. Green AgNPs synthesized with plant materials have been reported to have applications in the field of medicine due to their physicochemical properties. Previous works have also reported promising results for the application of green synthesized AgNPs as an anti-microbial agent for antibiotic-resistant bacterial strains, as well as for its cytotoxicity against cancer and tumoral cells, and for its improvement of drug delivery, dye degrading, and anticoagulant and antioxidant activity [6,7,8,9]. Although AgNPs are widely used worldwide due to their promising properties and effects, their increased production leads to a higher exposure rate to these nanoparticles which may cause adverse effects on health, thus, there is a necessity for safety evaluation of these green AgNPs [10]. Exposure to AgNPs can occur through inhalation, orally, or by dermal uptake. Regarding inhalation exposure, the distribution of AgNPs to the lungs induces inflammatory cell infiltration and chronic alveolar inflammation; after inhalation, these nanoparticles might be deposited in olfactory mucosa and later moved to olfactory nerves, which may cause neurotoxicity and immunotoxicity [11]. As for oral exposure to AgNPs, the contact of AgNPs with the acidic gastrointestinal environment helps the dissolution of AgNPs into silver ions; however, nanoparticles still exhibit size-dependent transition through the digestive system into the bloodstream [10]. In addition, reports show that AgNPs can penetrate healthy human skin and diffuse into the underlying structures, which may cause local reactions or systemic poisoning of the individual [12]. 

Despite the AgNPs current use and applications, there is also a gap in silver nanoparticle regulations. Regulatory institutions such as the European Chemicals Agency (ECHA), European Food Safety Authority (EFSA), European Medicines Agency (EMA), U.S. Food and Drug Administration (FDA), and the U.S. Environmental Protection Agency (EPA) are still working on defining the regulatory status of nanomaterials [13]. The permissible exposure to any form of silver is 0.01 mg/m^3^ (according to the National Institute for Occupational Safety and Health). However, to set up the regulation of AgNPs-based products, it is imperative to consider the type of product, purpose, and the exposure type it will have on humans and the environment; the FDA recommends detecting, quantifying, and characterizing these nanoparticles in products to regulate each of them, since nanomaterials are susceptible to batch-to-batch variation in their physicochemical properties [14,15]. In this matter, the regulation of AgNPs is necessary to develop new products and to avoid health and environmental hazards; however, the toxicity and adverse effects of AgNPs have not been thoroughly studied, and it is still necessary to evaluate their impact in an in vitro and in vivo manner [16,17]. 

In addition, there are a few gaps in the AgNPs research. First, the properties of AgNPs are directly related to the synthesis method used, as well as the particle size [18]; the size of the nanoparticles plays an important role in the toxicity of AgNPs, and research has not revealed how their properties (both physicochemical and morphological) influence their interaction with different biological systems (cells, tissues, and living organisms) and environments; the interactions of nanoparticles with biological systems is highly complex, so it is necessary to evaluate the mechanisms of toxicological activity at the cellular and molecular level, along with their neurotoxicity and immunotoxicity [17]. Previously, we developed green synthesized AgNPs using an aqueous peel extract of *S. queretaroensis*; this was the first report of a green synthesis of AgNPs based on this plant extract, which has a high betalain content that confers both antioxidant and anti-inflammatory properties. Furthermore, *S. queretaroensis* peel contains ascorbic acid (vitamin C) and reducing sugars that have an important role in silver reduction for the green synthesis of nanoparticles. The produced *S. queretaroensis* based AgNPs were characterized by ultraviolet visible spectroscopy (UV-Vis), and dynamic light scattering (DLS) analysis determined a size distribution between 20 and 600 nm, with an average particle size of 98.96 nm. We also assessed their antimicrobial activity, highlighting that the results were promising against fungi, Gram-negative and Gram-positive bacteria, and a methicillin-resistant strain of *Staphylococcus aureus*, which makes them an attractive option to be used in antimicrobial products; however, in order to further use these AgNPs as part of a product formulation, their safety needs to be assessed [5]. 

In this work, we aimed to perform extensive analysis on the toxicological effects of these *S. queretaroensis* green synthesized AgNPs at the different biological levels (cell, tissue, and organism) in both in vitro and in vivo systems. Particularly, we evaluated their cytotoxic effects using in vitro models to determine their toxic potential at a cellular level. Furthermore, we evaluated their toxic effects in animal models through different exposure pathways such as oral, dermal, and inhalation. We evaluated the toxic capacity of these *S. queretaroensis* AgNPs through several assays, including cell cytotoxicity, oral and dermal acute toxicity, dermal and ocular irritation, mutagenicity, genotoxicity, and inhalation toxicity, according to the OECD guidelines.

## 2. Materials and Methods

### 2.1. Plants, Microorganisms, Cell Lines, and Chemicals

The fruit of *Stenocereus queretaroensis* was acquired at a market in Guadalajara City. The bacterial strains TA 100 and TA 98 were obtained from Molecular Toxicology Inc. (North Carolina, USA). The L929 cell line was acquired from the American Type Culture Collection (ATCC), Virginia, USA. The animals used were Wistar rats, BALB/c mice, and New Zealand rabbits acquired from Bioterio Morelos (Morelos, Mexico). Triton X-100 rea-gents, 3-(4,5-dimethylthiazol-2-yl)-2,5-diphenyltetrazolium bromide (MTT), histidine, biotin, and Vogel-Bonner (VB) salts were acquired from Sigma-Aldrich (Burlington, MA, USA). Dimethylsulfoxide (DMSO), sodium chloride (NaCl), sodium hydroxide (NaOH), and glucose reagents were acquired from KARAL (Guanajuato, Mexico), while the reagents used for the cytotoxicity tests were acquired from Thermo Fisher Scientific (Waltham, MA, USA). Giemsa Kit reagents and immersion oil were purchased from HYCEL (Jalisco, Mexico). Agar reagents, sodium azide, 2-nitrofluorene, silver nitrate, and sodium pentobarbital were obtained from BD Bioxon (Bergen County, NJ, USA), Affymetrix (Santa Clara, CA, USA), and Fermont (Nuevo Leon, Mexico), respectively.

### 2.2. Silver Nanoparticles Synthesis

We had previously produced AgNPs with an aqueous *S. queretaroensis* peel extract. AgNPs were synthesized by adding the aqueous *S. queretaroensis* peel extract to a 2mM silver nitrate (AgNO_3_) solution at a volume ratio of 1:20. The pH was adjusted to 8 using 1M NaOH with an Orion 3 Star potentiometer Thermo Scientific) (Waltham, MA, USA). The mixture was stirred for 30 min at a temperature of 90 °C using a Cimarec stirring hot plate Thermo Scientific (Waltham, MA, USA), allowed to cool, and subsequently left to and stand for 24 h at room temperature to end the reaction. Then, the AgNPs solution was centrifuged at 15,000× *g* rpm for 20 min at 4 °C in a Sorvall Legend XTR centrifuge (Thermo Scientific, Waltham, MA, USA), and the supernatant was discarded. Finally, the pellet was washed with distilled water five times and once in 90% ethanol to obtain pure *S. queretaroensis*-mediated AgNPs [15].

### 2.3. In Vitro Assays

#### 2.3.1. Cell Culture and Cell Cytotoxicity Evaluation

Cell cytotoxicity was assessed with the L929 mouse fibroblast cell line, obtained from ATCC. L929 cells were cultured with Dulbecco’s Modified Eagle Medium (DMEM), 10% Fetal bovine serum (FBS), and 1% penicillin-streptomycin.

After reaching a 95% confluence in a T75 culture flask Falcon, Thermo Scientific (Waltham, MA, USA), cells were detached and seeded in 96-well plates Corning (Corning, NY, USA) of 10,000 cells/well. The culture media were changed 24 h after seeding, and the AgNPs extract was added to the culture media to reach the specific concentrations of 10, 20, 30, 50, 60, 80, and 500 µg/mL to test the cytotoxicity at different doses. Simultaneously, positive cytotoxicity controls were performed using TritonX-100 at different concentrations (0.001, 0.01, 0.17, and 3.21 µg/mL). All tests were performed in triplicate.

The cytotoxicity of AgNPs was determined using the MTT 3-(4,5-dimethylthiazol-2-yl)-2,5-diphenyltetrazolium bromide) method. Next, 24 h after treatment exposure, cells were washed twice with 1X Dulbecco’s phosphate-buffered saline (DPBS), a 0.5 mg/mL MTT solution was added to cell cultures, and cultures were incubated at 37 °C 5% CO_2_ for 4 h. Later, the MTT solution was discarded, and plates were left to dry at room temperature overnight. Finally, 100 µL of dimethyl sulfoxide (DMSO) was added to each well and the wells were incubated at room temperature under continuous agitation for an hour. Plates were read using a multi-well reader spectrometer model X-mark BIORAD (Hercules, CA, USA) at 570 nm. IC_50_ was calculated using the Statgraphics XVI.I software.

#### 2.3.2. Mutagenicity of AgNPs 

The mutagenicity of the AgNPs was assessed through the bacterial reverse mutation, or Ames, test under the OECD 471 guideline [19]. A pre-inoculum of the strains TA100 and TA98 was prepared (24 h, 100 rpm, 37 °C), then 100 µL of each inoculum strain was exposed to different concentrations of the AgNPs (0.039, 0.0195, 0.00975, 0.004875, and 0.0024375 µg/plate) for 20 min in 2 mL of top agar (agar, NaCl, Histidine/Biotin solution 0.5 mM), to be incorporated later in minimal glucose agar (MGA) plates (agar, vogel-bonner 50× salt solution, glucose solution 40% *v*/*v*). Cultures were incubated for 48 h at 37 °C, and two positive controls were used: Sodium azide (5 µg/plate), and 2-nitrofluorene (10 µg/plate) for TA100 and TA98, respectively.

### 2.4. In Vivo Evaluations

All the animals were handled according to the guidelines and regulations promulgated by the Federal Government of Mexico NOM-062-ZOO-1999 [20]. Animals were housed in polypropylene plastic cages at 23.0 ± 2.0 °C at 44–55% relative humidity (RH) and light and dark cycles of 12 h, with rodent food and water available ad libitum. All the protocols for experimental procedures were approved by the Internal Committee for the Care and Use of Laboratory Animals (ID: 2021-002A).

#### 2.4.1. Genotoxicity Evaluation

Two weeks before the experiment, 12 male Wistar rats (±180–200 g) were housed in cages under a twelve-hour light/dark cycle at 22 ± 3 °C and 60% humidity; they were fed with a standard diet, and their weight was recorded. Rats were randomly distributed into two groups (n = 6). AgNPs were orally administered daily (at 20 mg/mL for 60 days) to evaluate the genotoxic potential following the in vivo micronucleus test according to the 474 OECD guideline and the methods of Hayashi et al. (2016) [21,22]. Blood smears were made in triplicate on previously cleaned and degreased slides. Subsequently, the slides were left to dry at room temperature for 1 h and then stained with Giemsa stain diluted in phosphate buffer pH 6.8 in a 20:1 ratio for 7 min; then, they were washed with distilled water and dried at room temperature. The slides were analyzed under a light microscope BS-T20 (BESTSCOPE, Beijing, China) at 100× with oil immersion. The genotoxicity was evaluated by quantifying the micronucleated polychromatic erythrocytes (MNPCE) in 3000 polychromatic erythrocytes (PCE). Data were analyzed using GraphPad Prism 8.0.1 software, and Dunnett’s post hoc test was used for genotoxicity analysis with a *p* ≤ 0.05.

#### 2.4.2. Acute Oral Toxicity

The acute oral toxicity was evaluated according to the 425 OECD guidelines [23]. Five BALB/c female mice (∼23 g) fasted 4 h before dosing. After fasting, the mice were orally administered 2000 mg/kg^−1^ of AgNPs with a stainless steel cannula. After administration, we observed the mice closely every 30 min for the first 4 h, after which we periodically monitored the mice every 24 h for 14 days. At the end of the test, the animals were weighed and sacrificed to analyze their organs (liver, kidneys, lungs, and heart) and to search for signs of toxicity or macroscopic changes.

#### 2.4.3. Acute Dermal Toxicity 

The acute dermal toxicity was evaluated in female Wistar rats (±250 g) following the OECD 402 guidelines [24]. First, the rats were shaved in the dorsal section in an area equivalent to 10% of the total body surface area, and we topically applied the three sequential doses (200, 1000, and 2000 mg/kg^−1^) to the shaved area using a micropipette. Next, the treated area was covered with a gauze dressing. The animals were observed at 30 min, 2 h, and 6 h, and then monitored once daily for 14 days after treatment. We weighed the rats on days 1, 7, and 14. Once the test period was completed, the rats were sacrificed with sodium pentobarbital (150 mg/kg^−1^), and a necropsy was performed to evaluate the state of the organs and their physical characteristics (liver, kidneys, lungs, and heart). All experiments were conducted in duplicate.

#### 2.4.4. Dermal Irritation

Dermal irritation was assessed on three New Zealand albino rabbits according to the OECD 404 guidelines [25]. The rabbits were shaved in the dorsal section (~6 cm²) and 0.5 mL of 20 mg/mL of AgNPs were applied to this area. The exposure lasted 4 h, and the rabbits were monitored for signs of erythema and edema at 24, 48, and 72 h. The skin irritation scores were established according to the nature and severity of the lesions.

#### 2.4.5. Ocular Irritation 

Eye irritation was evaluated according to the OECD 405 guidelines in three albino New Zealand rabbits [26]. We instilled 100 µL of the AgNPs solution (20 mg/mL) into the bottom right conjunctival sac and kept their eyelids together for 20 min. The left eye was used as a control, and the response of the ocular structures (cornea, iris, and conjunctiva) was evaluated at 24, 48, and 72 h after treatment.

#### 2.4.6. Acute Inhalation Toxicity

The acute inhalation toxicity of AgNPs was evaluated according to the OECD 403 guideline [27]. Six rats (three females and three males) were exposed to AgNPs at 5 mg/L for 4 h in a full-body exposure chamber. The mean aerodynamic mass diameter was determined during the test using cascade impactor equipment. After the exposure, the animals were returned to their cages and monitored for 14 days to search for signs of toxicity. At the end of the experiment, the rats were euthanized, and a necropsy was performed to determine macroscopic changes or adverse effects in their internal organs. 

## 3. Results

### 3.1. Cell Cytotoxicity

We evaluated the cytotoxic effect of the biogenic AgNPs on the L929 cells using the MTT test. We treated the cell line with AgNPs in concentrations ranging from 10 to 500 µg/mL and calculated the half-maximal inhibitory concentration (IC_50_) of both the AgNPs and the Triton X100 positive control; the IC_50_ values were 134.76 µg/mL and 85.59 µg/mL, respectively. Additionally, we analyzed the cell morphological changes in cultures after exposure to AgNPs (Figure 1) and found morphology changes in a dose-dependent manner. In concentrations above 200 µg/mL, cells start to shrink and lose their fibroblast-like structures.

### 3.2. Mutagenicity

We assessed the mutagenicity of these AgNPs according to the bacterial reverse mutation test in the TA98 and TA100 strains. In addition, we determined the number of spontaneous revertants (SR) that is, those colonies that spontaneously repair the mutation and are used as test control, after treatment (Table 1). Based on the method used, the criterion for determining a mutagenic substance is that it increases two-fold the number of revertant colonies. The application of these AgNPs at different concentrations did not achieve a two-fold increase in the number of the test revertant colonies in comparison to SR; on the other hand, the substances used as positive controls, sodium azide, and 2-nitrofluorene, resulted in a number of colonies that indicated a mutagenic effect. These results suggest that these AgNPs are not mutagenic at the tested concentrations. The results are presented as the mean and standard deviation.

### 3.3. Genotoxicity

We assessed the genotoxic potential of these AgNPs through the in vivo micronucleus test. We did not find any significant increase (*p* = 0.8037) in peripheral blood MNPCE in the test group that received AgNPs orally for 60 days in comparison to the negative control that received distilled water (DW). MNPCE indicated a clastogenic effect (Table 2). Data are expressed as means ± standard deviations of MNPCE/3000 PCE. In addition, Dunnett’s multiple post hoc analysis did not indicate statistically significant differences (*p* ≤ 0.05) in the test group vs. the control group (DW).

### 3.4. Acute Oral Toxicity

After oral exposure, according to the 425 OECD guidelines, animals were monitored closely for 14 days. During this period, we did not observe any toxicological effects, such as changes in the skin, coat, eyes, mucous membranes; behavioral patterns; or respiratory, circulatory, and nervous systems. The animals were weighed on days 0, 7, and 14 (Table 3). There was no mortality during the test. After the 14-day observation period, the mice were euthanized and a necropsy was performed to extract, analyze, and weigh the organs (liver, kidney, lungs, and heart). In addition, no morphologically or macroscopic changes were observed (Table 4).

### 3.5. Acute Dermal Toxicity 

After assessing the acute dermal toxicity, we monitored all groups carefully and did not observe any changes after the first 30 min, 2 h, and 6 h in the area exposed to the AgNPs. During the rest of the monitoring, we did not observe changes in the skin, coat, eyes, mucous membranes, behavior patterns, or the respiratory and circulatory systems. After the 14-day observation period, there was no mortality, and the animals were weighed on days 0, 7, and 14 (Table 5). On day 14, we euthanized the animals, performed a necropsy, and extracted the organs (liver, kidneys, lungs, and heart) to rule out macroscopic changes (Table 6). We did not observe any macroscopically relevant changes in the internal organs in comparison to the control group. These results suggest that *S. queretaroensis* AgNPs do not cause acute dermal toxicity at the tested concentrations (200, 1000, and 2000 mg/kg^−1^).

### 3.6. Dermal Irritation Test

We performed dermal irritation tests for these AgNPs in New Zealand male albino rabbits. We did not observe any severe dermal response (Table 7 and Figure 2). The skin response was scored according to the OECD 404 guidelines: mean skin response score = (total erythema and eschar formation score + total edema formation score)/3.

### 3.7. Ocular Irritation Test 

We evaluated the degree of eye irritation according to the OECD 405 guidelines by scoring the cornea, iris, and conjunctiva lesions at specific intervals after treatment. Irritation was graded on a scale from 0 to 4 (from the lowest to the highest level of irritation, where the number 4 stands for a severe response). The classification of irritation scores from 24 h is listed in Table 8; we did not observe any severe irritation response in the New Zealand rabbit model. There were no ocular lesions in the animals exposed to the solution of AgNPs (Figure 2).

### 3.8. Acute Inhalation Toxicity

The acute inhalation toxicity of the AgNPs was evaluated according to the OECD 403 guideline; the mean aerodynamic mass was determined to be 1.33 microns, with a geometric standard deviation (σg) of 2.17. This test did not produce any mortality 14 days after inhalation exposure, and no severe clinical signs were observed after the 14-day evaluation period (Table 9). At the end of the test period, we evaluated the internal organs of the test animals, and there were no relevant macroscopic changes (Table 10). The weights of tested animals were recorded on days 0, 7, and 14 (Table 11). The administration of these AgNPs through inhalation uptake did not show lethality, so the LC_50_ is greater than 5 mg/L.

## 4. Discussion 

Lately, there has been an increase in the usage of AgNPs in consumer and scientific applications, such as food packaging, antimicrobial agents, textile and pharmaceutical production, etc. However, there is still a lack of information on their nano-toxic potential. Previously, we had developed AgNPs based on an *S. queretaroensis*, and these showed a high antimicrobial potential; however, in order to further explore their application, it is necessary to assess their toxicologic activity in biologic systems. In this work, we evaluated the toxicological effects of *S. queretaroensis* AgNPs using in vitro and in vivo models.

Additionally, we performed an Ames mutagenicity assay, the gold-standard method for determining the mutagenic potential of chemical substances such as plants or products for initial testing [28]. We tested the mutagenic potential of our green synthesized AgNPs against the TA98 and TA100 strains. The results reflect that these AgNPs are not mutagenic up to 0.03 µg/plate. This takes into account the fact that any substance can be considered a mutagen if it demonstrates at least a two-fold concentration-dependent increase in the mean revertant colonies per plate in any one of the tested strains [29]; these AgNPs did not induce any large-scale DNA damage that could be detected by the Ames test. The results are consistent with previous studies that evaluate the mutagenic potential of biogenic AgNPs. Several studies assess the mutagenicity of green synthesized AgNPs concentrations ranging from 0.15 µg/plate to 0.5 mg/ plate without any mutagenic responses towards the tested strains, including the TA98 and TA100 [30,31]. On the other hand, previous reports have found that the antimicrobial activity of AgNPs inhibits bacterial growth, which may reduce the sensitivity of the test; Xiaoqing et al., (2016) performed a cell uptake experiment that showed that AgNPs were not internalized by the bacterial cells, as the bacterial cell wall is a barrier that prevents nanoparticles from entering the cells [32]. This opposite outcome in the Ames test suggests that it is important to further extend the mutagenic bacterial tests to mammalian cell models.

We also evaluated the genotoxicity of these AgNPs and determined their cytogenetic damage potential by quantifying the formation of micronuclei in vivo, which is the result of chromosomal damage in the erythroblasts of the test individual. Our results show no significant difference between the peripheral blood MNPCE of the test and the control groups (1.62 ± 1.32 test group against 1.44 ± 0.81 control). Similarly, Kim et al. (2008) evaluated genotoxicity after a 28-day oral exposure (at 0, 30, 300, and 1000 mg/kg/day) [33]. Kim et al. (2008) and Y. S. Kim et al. (2008) did not report any significant differences between the MNPCE/2000 PCE ratio of the test group vs. control group (3.5, 2.40, and 3.4), thus showing an absence of genotoxicity. Moreover, Kim et al. (2011) reported a lack of biogenic AgNPs genotoxicity after a 90-day inhalation exposure assay with doses of 30, 300, and 1000 mg/kg/day, since the resulting rats did not exhibit any effect on the frequency of their micronucleated polychromatic erythrocytes ([34].

We separately assessed the acute toxicity of biogenic silver nanoparticles through oral and dermal exposure according to the OECD 425 and 402 guidelines. These evaluations are important to determine the biological risks of substances such as biogenic AgNPs. After 14 days of oral exposure to AgNPs (2000 mg/kg^−1^), we did not observe any changes in physiological and conduct patterns reflecting any signs of toxicity, and there was no significant difference between initial and final body weights; this suggests that the half-lethal dose (LD_50_) is greater than 2000 mg/kg; as for acute dermal toxicity, we did not find any signs of physiologically adverse effects nor significant difference in body weights; in fact, both experiments were successfully carried out without the deaths of any individuals. Alwan et al. (2021) reported the safety of the oral administration of biogenic AgNPs; their results did not show any difference in body weights or the health state of mice, and there were no changes in biochemical blood and histopathological parameters [35]. However, another report suggests that the oral administration of AgNPs leads to accumulation of these nanoparticles in major organs such as the liver, kidneys, spleen, and brain due to repeated oral exposure. Variations in outcomes relied on the silver nanoparticle size, synthesis method, time of exposure, and concentration of AgNPs administered [36].

Furthermore, we also assessed the ocular and dermal irritation of these AgNPs, and the results showed that there was no mortality in the outcome; according to the guidelines, it is important to humanely euthanize test animals if they show any signs of severe eye lesions, distress, and/or pain at any stage of the observation period, and there was no need to euthanize any of the test subjects. Moreover, there were no signs of irritation or corrosion reactions on the eye structures or the skin; this is consistent with a previous study regarding the safety evaluation of green synthesized AgNPs. In addition, we did not find any mortality signs after inhalation exposure, and the results are consistent with Hyun et al. (2008). After continuous inhalation exposure to AgNPs, there were not any relevant changes or statistical differences compared to the control groups [37]. However, after repeated five-day inhalation exposure, the researchers in [38] found an increase in oxidative stress associated with inflammation. These results suggest that it is important to further evaluate the adverse effects of AgNPs inhalation after long-term exposure. To summarize our findings and compare our results with previous safety assessments, we created a comparative profile, shown in Table 12.

## 5. Conclusions and Future Perspectives

In conclusion, we evaluated the cytotoxicity, genotoxicity, and mutagenic potential, and also performed acute oral and dermal toxicity tests, to determine the toxic effects of green synthesized AgNPs based on *S. queretaroensis* peel extract, as well as skin and ocular irritation following the OECD test guidelines. We observed a dose-dependent manner of cytotoxicity for AgNPs (IC_50_ 134.76 µg/mL). The green synthesized *S. queretaroensis* did not exhibit mutagenic damage in the Ames test in a concentration up to 0.03 µg/plate. Acute toxicity through oral and dermal exposure of AgNPs to in vivo models was not detected during the 14-day trial period at 2000 mg/kg doses. There was no sign of adverse reactions after ocular and dermal irritation assessments. The testing of green *S.queretaroensis*-based AgNPs did not show any adverse effects from its application in in vitro and in vivo models, which would allow these nanoparticles to be used as part of an antimicrobial or therapeutic product; however, these evaluations were short-term, and the effects of AgNPs on full biological systems under long-term exposure have not yet been addressed. It is necessary to further evaluate the interactions between AgNPs and complex biological systems in a time-dose dependent manner.

## Figures and Tables

**Figure 1 materials-15-05700-f001:**
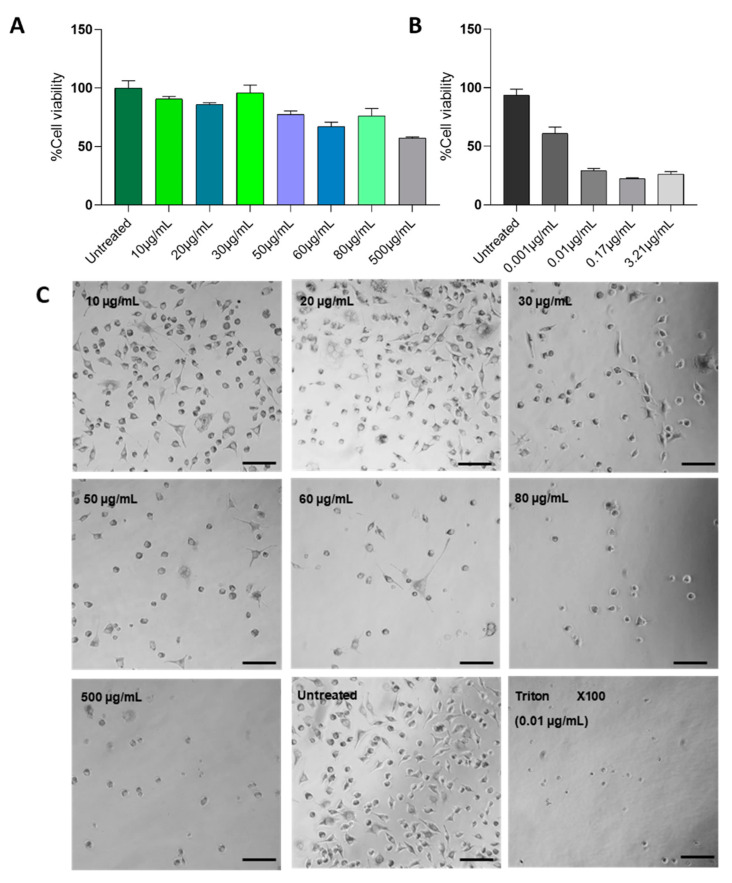
(**A**) Cell viability MTT assay results of the in vitro cytotoxicity of AgNPs against L929 cells for 24 h. (**B**) Triton X100 positive cytotoxicity control. Data are expressed as mean ± SD of three experiments. The percentage of cytotoxicity is expressed relative to untreated controls (**A**,**B**). (**C**) Phase-contrast micrographs of in vitro cell cytotoxicity assessment of *S. queretaroensis*-based AgNPs. Scale bar: 200 µM.

**Figure 2 materials-15-05700-f002:**
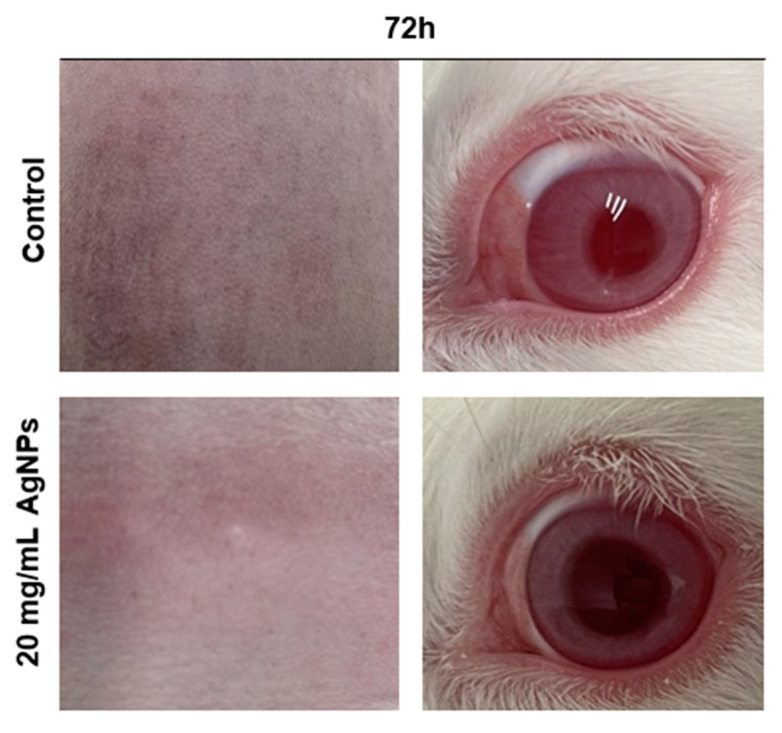
Rabbit dermal and ocular response 72 h after 20 mg/mL AgNPs treatment.

**Table 1 materials-15-05700-t001:** Revertant colonies obtained in the Ames test of AgNPs.

Sample (µg/Plate)	TA100 (Colonies)	TA98 (Colonies)
0.039	78 ± 1.41	12.5 ± 0.70
0.0195	68 ± 5.66	15.5 ± 2.12
0.00975	84 ± 2.83	17 ± 1.41
0.004875	80 ± 2.83	18 ± 1.41
0.0024375	104 ± 2.83	20 ± 1.41
SR	89.5 ± 9.19	21 ± 2.83
NaN_3_ (5 µg/plate)	398 ± 36.77	-
2-Nitrofluorene (10 µg/plate)	-	74.5 ± 3.53

Results are presented as the mean plus standard deviation.

**Table 2 materials-15-05700-t002:** Micronucleated polychromatic erythrocytes (MNPCE) in 3000 polychromatic erythrocytes (PCE) from peripheral blood.

Group	n	MNPCE/ 3000 PCE	*p* ≤ 0.05
DW	6	1.44 ± 0.81	NO
AgNPs	6	1.62 ± 1.32	NO

Results are presented as the mean plus standard deviation.

**Table 3 materials-15-05700-t003:** Weight (g) of the animals orally exposed to the AgNPs.

Animal	Doses (mg/kg^−1^)	Day 0	Day 7	Day 14
Mouse 1	2000	20.41	20.4	21.28
Mouse 2	2000	20.60	20.71	21.04
Mouse 3	2000	20.50	20.56	20.72
Mouse 4	2000	20.30	22.21	22.89
Mouse 5	2000	20.20	20.87	21.80

**Table 4 materials-15-05700-t004:** Weight of the organs of the animals orally exposed to the AgNPs.

Animal	Doses (mg/kg^−1^)	Liver (g)	Left Kidney (g)	Right Kidney (g)	Lungs (g)	Heart (g)
Mouse 1	2000	0.95	0.13	0.14	0.19	0.12
Mouse 2	2000	1.10	0.15	0.15	0.21	0.12
Mouse 3	2000	0.91	0.13	0.13	0.18	0.11
Mouse 4	2000	1.03	0.12	0.13	0.18	0.13
Mouse 5	2000	1.11	0.14	0.15	0.22	0.12

**Table 5 materials-15-05700-t005:** Weight (g) of the animals dermally exposed to the AgNPs.

Animal	Doses (mg/kg^−1^)	Day 0	Day 7	Day 14
Rat 1	200	261.5	277.4	271.4
Rat 2	200	247.3	259.9	265.4
Rat 3	1000	263.2	268.3	270.8
Rat 4	1000	259.8	278.8	282.2
Rat 5	2000	256.1	266.5	274.5
Rat 6	2000	254.3	264.9	271.5

**Table 6 materials-15-05700-t006:** Weight of the organs of the animals dermally exposed to the AgNPs.

Animal	Doses (mg/kg^−1^)	Liver (g)	Left Kidney (g)	Right Kidney (g)	Lungs (g)	Heart (g)
Rat 1	200	8.21	0.92	0.91	1.39	1.08
Rat 2	200	9.50	0.96	1.07	1.29	1.03
Rat 3	1000	9.40	0.82	0.80	1.27	1.04
Rat 4	1000	9.52	0.98	1.05	1.47	1.02
Rat 5	2000	8.62	0.82	0.92	1.71	0.86
Rat 6	2000	8.14	0.80	0.85	1.38	0.87

**Table 7 materials-15-05700-t007:** Rabbit dermal response after AgNPs treatment.

	1 h	24 h	48 h	72 h	Mean
**Test repetition**			**Erythema**		
Rabbit 1	0	0	0	0	0.00
Rabbit 2	1	1	0	0	0.33
Rabbit 3	1	1	0	0	0.33
	**Edema**				
Rabbit 1	0	0	0	0	0.00
Rabbit 2	1	0	0	0	0.00
Rabbit 3	1	0	0	0	0.00

**Table 8 materials-15-05700-t008:** Classification of ocular lesions in New Zealand rabbits after AgNPs treatment.

Time After Treatment	Cornea	Iris	Conjunctiva (Redness)	Conjunctiva (Chemosis)
**Rabbit #1**				
Hour 1	1	0	2	0
Hour 24	0	0	1	0
Hour 48	0	0	0	0
Hour 72	0	0	0	0
**Average test score**	0.00	0	0.33	0
**Rabbit #2**				
Hour 1	1	0	2	0
Hour 24	1	0	1	0
Hour 48	0	0	1	0
Hour 72	0	0	0	0
**Average test score**	0.33	0.00	0.66	0.00
**Rabbit #3**				
Hour 1	1	0	1	0
Hour 24	0	0	1	0
Hour 48	0	0	0	0
Hour 72	0	0	0	0
**Average test score**	0.00	0.00	0.33	0.00

**Table 9 materials-15-05700-t009:** Clinical results of test individuals after AgNPs inhalation exposure.

Test ID	Hours after Exposure					Days after Exposure													
	0.5	1	2	3	4	1	2	3	4	5	6	7	8	9	10	11	12	13	14
F1	+	+	+	+	+	+	+	+	+	+	+	+	+	+	+	+	+	+	+
F2	+	+	+	+	+	+	+	+	+	+	+	+	+	+	+	+	+	+	+
F3	+	+	+	+	+	+	+	+	+	+	+	+	+	+	+	+	+	+	+
M1	+	+	+	+	+	+	+	+	+	+	+	+	+	+	+	+	+	+	+
M2	+	+	+	+	+	+	+	+	+	+	+	+	+	+	+	+	+	+	+
M3	+	+	+	+	+	+	+	+	+	+	+	+	+	+	+	+	+	+	+

F: female rat; M: male rat; 1–3: animal number; +: normal, no changes detected.

**Table 10 materials-15-05700-t010:** Macroscopical evaluation of internal organs after acute inhalation toxicity testing.

	Test ID					
Organs	F1	F2	F3	M1	M2	M3
Stomach	+	+	+	+	+	+
Bowels	+	+	+	+	+	+
Lungs	+	+	+	+	+	+
Liver	+	+	+	+	+	+
Heart	+	+	+	+	+	+
Spleen	+	+	+	+	+	+

F: female rat; M: male rat; 1–3: animal number; +: normal, no changes detected.

**Table 11 materials-15-05700-t011:** Weight of animals after acute inhalation toxicity testing.

Test ID	Weight (g)		
	Day 0	Day 7	Day 14
F1	220	226	233
F2	216	220	224
F3	221	226	232
M1	218	222	228
M2	213	217	222
M3	215	220	224

F: female rat; M: male rat; 1–3: animal number; +: normal, no changes detected.

**Table 12 materials-15-05700-t012:** Previous findings on the safety assessment of silver nanoparticles.

Assesment	Model	AgNPs Synthesis Source	Findings and Tested Doses	Citation
Cytotoxicity	L929 mouse fibroblasts	Dextrose based synthesis	IC_50_ = 168.47 µg/mL	[39]
	T47D human breast cancer cells	*Rosa damascena* plant extract	IC_50_ = 6.31 µg/ml	[40]
	L929 mouse fibroblasts	Actinobacterial (NH28 strain) synthesis	64.5 μg/mL	[41]
Mutagenicity (Ames)	TA98, TA100, TA1535, TA1537	*Streptomyces griseorubens* (AU2 strain)	No mutagenic effect from 0.01 µg/mL to 10 µg/mL.	[31]
	TA98 and TA100	Ag-NPs (Sigma-Aldrich)	No mutagenic effect at 250, 100, and 50 µg/plate.	[30]
	TA98 and TA100	BioPure^TM^ AgNPs from Nano-Composix Inc.	Nonconclusive.	[32]
	TA100, TA102, and TA1535	AgNPs from Novacentrix Inc.	No mutagenic effect from 0.15 to 76.8 µg/plate.	[42]
	TA98	Actinobacterial (SH11) synthesized AgNPs	No mutagenic effect from 0.15–100 µg/plate. Inhibitory activity of bacterial growth.	[43]
Genotoxicity	Rat	AgNPs from Namatech Co.	No significant genotoxic effect at 0, 30, 300, and 1000 mg/kg/day.	[33]
	Rat	AgNPs from Namatech Co.	No significant gnotoxic effect at 0, 30, 300, and 1000 mg/kg/day.	[34]
Acute oral toxicity	Rat	*C. zeylanicum* based synthesis	No mortality; no weight changes. No signs of toxicity; no physiological nor histopatological changes detected(0.85, 1.76 and 3.53 mg/kg doses).	[35]
	Mice	AgNPs from BioPure^TM^	No treatment-related clinical signs or mortality; no body and organ weights effects. Treatment-related changes in hematology and clinical chemistry after recovery (0.25 and 1 mg AgNPs).	[36]
Eye irritation	Rabbit and guinea pigs	AgNPs from Namatech Co.	No significant clinical signs nor mortality, and no acute irritation or corrosion reaction for the eyes and skin. One guinea pig showed discrete erythema;	[34]
	Guinea pigg	Chemical reduction	AgNPs at 5000 ppm produced transient eye irritation during early 24 h observation time.	[44]
Acute inhalation toxicity	Rat	AgNPs from Namatech Co.	Silver nanoparticles have an influence on the neutral mucins in the respiratory mucosa, without toxicological significance.	[38]

## Data Availability

The data presented in this study are available on request to the corresponding author.

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
