# Peer review of "Toxicological Evaluation of Silver Nanoparticles Synthesized with Peel Extract of Stenocereus queretaroensis"

_materials, 2022, doi:10.3390/ma15165700_

Round 1

Reviewer 1 Report

The manuscript is focused on the evaluation of the toxicological effects of silver nanoparticles biosynthesized from Stenocereus queretaroensis peel extracts. 

The introduction is well written, and the purpose of the study is clearly presented: assessing the safety of AgNPs through in vitro and in vivo methods.

The methods and results are clearly presented and some dose dependent effects were observed on cell viability for the in vitro assays. 

The in vivo evaluation of the AgNPs did not exhibit severe adverse effects, however long-therm exposure studies are stiil needed.

Author Response

Thank you for your comments. In addition, we have made an English editing service of MDPI.

Reviewer 2 Report

The current study is evaluating the toxicity of silver nanoparticles (NPs) that were synthesized using a peel extract of Stenocereus queretaroensisby different methods including cytotoxicity, genotoxicity and in vivo in Rabbits. It is important to evaluate the toxicity of the silver nanoparticles to avoid them being harmful to mammalian cells. Silver nanoparticles are used in several applications, and it is a growing field, and that using natural products in the synthesis steps, is also a growing field. The aim was clear and important, however, highlighting the impact of the study in more details was lacking, as well as proofreading the English style, to avoid the conversational style and to be more precise in the wording and with more details.

Major points:

1-      The term ‘biosynthesis’ is confusing because it is mainly used if the organism itself is synthesizing the compound, so to avoid confusion, it is best to elaborate on the term and clarifying it. 

2-      Line 21 in the abstract ‘several’, more details need to be added, as to the specific methods used.

3-      Page 2, lines 50 and 51, the sentence requires re-wording, as to the English and for more details to be added, the whole paragraph needs elaboration with more details, as it is rather short.

4-      Page 2 line 68: ‘organisms’ requires re-wording, is it meant ‘regulatory institutions’?

5-      Page 2 line 83, ‘visually’, is it a qualitative or quantitative test? It requires further elaboration.

6-      Page 4, line 152, the sentence seems to be truncated and needs to be revised for completion and clarification.

7-      Page 6, line 244, the spontaneous RE mutants requires clarification.

8-      Page 6, line 245, that there was not a two-fold increase needs clarification as to why this is important to mention, to avoid confusion.

9-      Page 6, line 246, the exact positive controls and details need to be mentioned in more details.

10-   Table 1, the columns of TA100 and TA98, the numbers are not clearly understood from the table, it is better if they are written in the legend of the table.

11-   Page 6, line 254, ‘EPMN’, is not written in full upon the first appearance in the text, and it is best to explain its relevance here.

12-   Table 4, the columns with no unit are confusing to be directly understood.

13-   Table 5, ‘the animals’, more details need to be written about them in the table, are they the rabbits?!

14-   Page 11, line 368, a conversational style is best to be avoided and re-wording is required.

15-   The last paragraph in the discussion requires re-writing as it is lacking specific details and some parts were rather vague or require grammatical proofreading such as ‘the results didn’t exhibit mortality’, requires re-wording and more details.

Minor points:

1-             The scientific name of the organism should be italicized throughout the text e.g. in the abstract: Stenocereus queretaroensis.

2-             Line 26: ‘these AgNPs’, the word needs to be clarified, is it referring to those synthesized by this particular method?!

3-             The formatting of the manuscript needs proofreading, as sometimes there are spaces at the beginning of each paragraph and sometimes there is no space.

4-             Missing reference: Page 1, Line 46.

5-             Page 2 line 57, ‘get contact’, requires re-wording or proofreading. And in general, the whole manuscript requires grammatical proofreading.

6-             Page 2, line 72, ‘m3’ to be ‘m3’. Same for ‘CO2’ in page 3, line 121. Line 148, for the degree sign not to be underlined.

7-             Figure 1A, the micro symbol on the graph needs to be corrected, as it is not the correct one at the moment. And Figure 1B, on the graph axis labels, ‘.001’, can it be ‘0.001’ to avoid confusion or ‘10*10-4’ and revise for consistency.

8-             Proofreading is required, in page 7, line 265, ‘weighted’, and also to use the past passive tense throughout the text.

Author Response

Major points:

  • The term ‘biosynthesis’ is confusing because it is mainly used if the organism itself is synthesizing the compound, so to avoid confusion, it is best to elaborate on the term and clarifying it. 

Response: The term “biosynthesis” was change in all text including title

Line 21 in the abstract ‘several’, more details need to be added, as to the specific methods used.

Response: We included details of methods in abstract

  • Page 2, lines 50 and 51, the sentence requires re-wording, as to the English and for more details to be added, the whole paragraph needs elaboration with more details, as it is rather short.

Response: Introduction has been rewritten

  • Page 2 line 68: ‘organisms’ requires re-wording, is it meant ‘regulatory institutions’?

Response: The word “organisms” was replaced by “institutions”

  • Page 2 line 83, ‘visually’, is it a qualitative or quantitative test? It requires further elaboration.

Response: The word “visually” was deleted and phrase was complemented

  • Page 4, line 152, the sentence seems to be truncated and needs to be revised for completion and clarification.

This comment has been addressed, the missing complement was added to the sentence

7-      Page 6, line 244, the spontaneous RE mutants requires clarification.

Response: The term “spontaneous mutants” was clarified

8-      Page 6, line 245, that there was not a two-fold increase needs clarification as to why this is important to mention, to avoid confusion.

Response: The term “two-fold” was explained

9-      Page 6, line 246, the exact positive controls and details need to be mentioned in more details.

Response: The positive controls used in mutagenicity test were detailed

10-   Table 1, the columns of TA100 and TA98, the numbers are not clearly understood from the table, it is better if they are written in the legend of the table.

Response: We added units in columns on TA100 and TA98. Also, we added the legend “Results are presented as the mean and standard deviation”

11-   Page 6, line 254, ‘EPMN’, is not written in full upon the first appearance in the text, and it is best to explain its relevance here.

Response: The abbreviation “EPMN” was changed to “MNPCE” it was written in full in its first appearance, and homogenized in all text

12-   Table 4, the columns with no unit are confusing to be directly understood.

Response: We added units in columns of Tables in all document

13-   Table 5, ‘the animals’, more details need to be written about them in the table, are they the rabbits?!

Response: We added the type of animals used in all Tables

14-   Page 11, line 368, a conversational style is best to be avoided and re-wording is required.

This paragraph has been rewritten in a more formal style.

15-   The last paragraph in the discussion requires re-writing as it is lacking specific details and some parts were rather vague or require grammatical proofreading such as ‘the results didn’t exhibit mortality’, requires re-wording and more details.

Last paragraph was re-written and details were provided to clarify the “mortality” term used.

Minor points: 

1-             The scientific name of the organism should be italicized throughout the text e.g. in the abstract: Stenocereus queretaroensis.

Response: All scientific names in manuscript were italicized

2-             Line 26: ‘these AgNPs’, the word needs to be clarified, is it referring to those synthesized by this particular method?!

On line 26, we clarified the subject of our sentence. We were referring to our previously synthesized S.queretaroensis AgNPs

3-             The formatting of the manuscript needs proofreading, as sometimes there are spaces at the beginning of each paragraph and sometimes there is no space.

We used the MDPI english editing service

4-             Missing reference: Page 1, Line 46.

Missing reference was added to page 1, Line 46. However, since part of the text was rewritten above, this is not on line 46.

5-             Page 2 line 57, ‘get contact’, requires re-wording or proofreading. And in general, the whole manuscript requires grammatical proofreading.

Phrasal verb and preposition usage were corrected

  • Page 2, line 72, ‘m3’ to be ‘m3’. Same for ‘CO2’ in page 3, line 121. Line 148, for the degree sign not to be underlined.

Response: All signs were reviewed and corrected in manuscript

7-             Figure 1A, the micro symbol on the graph needs to be corrected, as it is not the correct one at the moment. And Figure 1B, on the graph axis labels, ‘.001’, can it be ‘0.001’ to avoid confusion or ‘10*10-4’ and revise for consistency.

The micro symbol on the graph has been corrected, as well as the axis on Figure 1B.

8-             Proofreading is required, in page 7, line 265, ‘weighted’, and also to use the past passive tense throughout the text.

Complete paragraph of acute oral toxicity was re-written in past passive tense throughout the text

Reviewer 3 Report

I reviewed the article with ID =materials-1832687.  The article topic is intriguing and promising in the area. Overall, the article structure and content are suitable for the Materials journal. I am pleased to send you major level comments, there are some serious flaws which need to be corrected before publication. Please consider these suggestions as listed below.

1.      The title seems good, but the abstract seems to be wired. Please add one more introductory line of your objective in beginning of abstract.

2.      Keywords are ok

3.      Research gap should be delivered on more clear way with directed necessity for the future research work.

4.      Introduction section must be written on more quality way, i.e., more up-to-date references addressed. Please target the specific gap.

5.      The novelty of the work must be clearly addressed and discussed, compare previous research with existing research findings and highlight novelty.  

6.      What is the main challenge? Why author choose this material? Please highlight in the introduction part.

7.      Page 1 Line 39 need a reference please cite this one- Silver nanoparticles: various methods of synthesis, size affecting factors and their potential applications–a review.

8.      The main objective of the work must be written on the more clear and more concise way at the end of introduction section.  

9.      Line 46, Page 1 needs a string reference please cite this one- Recent advances in metal decorated nanomaterials and their various biological applications: a review.

10.   Overall, introduction need a sincere effort, it seems very weird.

11.   Please check the abbreviations of words throughout the article. All should be consistent.

12.   Please include all chemical/instrumentation brand name and other important specification.

13.   Please add chemical reagents section and stated all chemical with brand specifications.

14.   Regarding the replications, authors confirmed that replications of experiment were carried out. However, these results are not shown in the manuscript, how many replicated were carried out by experiment?

15.   Please see the table formatting. Its should be consistent.

16.   Please extend the Mutagenicity discussion.

17.   Please add a comparative profile section to compare your results and prove how it better than previous.

18.   Section 5 should be renamed by Conclusion and Future perspectives. Conclusion section is missing some perspective related to the future research work, quantify main research findings, highlight relevance of the work with respect to the field aspect. In present form chapter 5 is totally weird.

19.   To avoid grammar and linguistic mistakes, Major level English language should be thoroughly checked. Please revise your paper accordingly since several language issue occurs on several spots in the paper.

20.   Reference formatting need carefully revision. All must be consistent in one formate. Please follow the journal guidelines. The present style of reference is not MDPI style especially in the text. Please use [] this bracket.

Author Response

  1. The title seems good, but the abstract seems to be wired. Please add one more introductory line of your objective in beginning of abstract.

We have added one more introductory line of the objective in this work. The term “biosynthesized” was replaced to “synthesized” according to suggestion of reviewer 2.

  1. Keywords are ok

Response: Thank you for your comment

  1. Research gap should be delivered on more clear way with directed necessity for the future research work.

Overall, the introduction was rewritten adding the requested details about novelty and research gaps.

  1. Introduction section must be written on more quality way, i.e., more up-to-date references addressed. Please target the specific gap.

Overall, the introduction was rewritten adding the requested details and more references

  1. The novelty of the work must be clearly addressed and discussed, compare previous research with existing research findings and highlight novelty. 

Overall, the introduction was rewritten adding the requested details. We added a comparative table with previous research

  1. What is the main challenge? Why author choose this material? Please highlight in the introduction part.

Overall, the introduction was rewritten adding the requested details. We added an explanation of the properties of S. queretaroensis and details about the previously produced S.queretaroensis AgNPs.

  1. Page 1 Line 39 need a reference please cite this one- silver nanoparticles: various methods of synthesis, size affecting factors and their potential applications–a review.

We added the suggested citation

  1. The main objective of the work must be written on the more clear and more concise way at the end of introduction section. 

Overall, the introduction was rewritten adding the requested details

  1. Line 46, Page 1 needs a string reference please cite this one- Recent advances in metal decorated nanomaterials and their various biological applications: a review.

We added the suggested citation

  1. Overall, introduction need a sincere effort, it seems very weird.

We have worked on introduction adding relevant information

  1. Please check the abbreviations of words throughout the article. All should be consistent.

All abbreviations were reviewed and corrected when necessary

  1. Please include all chemical/instrumentation brand name and other important specification.

We included brand name of chemical and instruments used

  1. Please add chemical reagents section and stated all chemical with brand specifications.

We included section 2.1 “Plants, Microorganisms, Cell Lines, and Chemicals” in Materials and methods

  1. Regarding the replications, authors confirmed that replications of experiment were carried out. However, these results are not shown in the manuscript, how many replicated were carried out by experiment?

For in vitro assays “All tests were performed in triplicates” (line 215). For in vivo evaluations the number of animals used are the number of repetitions; for genotoxicity n=6, acute oral toxicity n=5, acute dermal toxicity n=6, dermal and ocular irritation test n=3, acute inhalation toxicity n=6

  1. Please see the table formatting. It should be consistent.

Tables in manuscript were checked and corrected

  1. Please extend the Mutagenicity discussion.

We have extended the mutagenicity discussion.

  1. Please add a comparative profile section to compare your results and prove how it better than previous.

We have added a table to compare our results with previous findings.

  1. Section 5 should be renamed by Conclusion and Future perspectives. Conclusion section is missing some perspective related to the future research work, quantify main research findings, highlight relevance of the work with respect to the field aspect. In present form chapter 5 is totally weird.

Section 5 was renamed to conclusions and future perspectives and Conclusion has been rewritten

  1. To avoid grammar and linguistic mistakes, Major level English language should be thoroughly checked. Please revise your paper accordingly since several language issue occurs on several spots in the paper.

We have used the English language editing by MDPI (see certificate in attachment).

  1. Reference formatting need carefully revision. All must be consistent in one formate. Please follow the journal guidelines. The present style of reference is not MDPI style especially in the text. Please use [] this bracket.

We have changed citation style to MDPI

Round 2

Reviewer 2 Report

The authors have addressed all the comments. 

Reviewer 3 Report

Accepted in current form